# The Microstructure and Conductivity Evolution of Plasma-Sprayed (Mn, Co)$_3$O$_4$ Spinel Coatings during Conductivity Measurements at Elevated Temperature

**Jianbo Zou** [1,2], **Chen Song** [2,*], **Kui Wen** [2], **Taikai Liu** [2,*], **Chunming Deng** [2], **Min Liu** [2] **and Chenghao Yang** [3]

1    School of Materials Science and Engineering, Central South University, Changsha 410083, China; zoujianbo_2018@163.com
2    National Engineering Laboratory for Modern Materials Surface Engineering Technology, The Key Lab of Guangdong for Modern Surface Engineering Technology, Guangdong Institute of New Materials, Guangdong Academy of Science, Guangzhou 510651, China; wenkui@gdinm.com (K.W.); dengchunming@gdinm.com (C.D.); liumin@gdinm.com (M.L.)
3    School of Environment and Energy, South China University of Technology, Guangzhou 510006, China; esyangc@scut.edu.cn
*    Correspondence: songchen@gdinm.com (C.S.); liutaikai@gdinm.com (T.L.)

**Abstract:** (Mn, Co)$_3$O$_4$ spinel is widely used to protect the metallic interconnect of solid oxide fuel cells while it suffers deoxidization during the preparation by plasma spray. This work was proposed to study the effect of spray parameters on the microstructure and conductivity of spinel coatings. In this work, spinel coatings were prepared by the atmospheric plasma spray. The prepared coatings were heated up to 700 °C and held on for 15 h to allow the conductivity evolution. The microstructure and composition of coatings were characterized by scanning electron microscopy (SEM), transmission electron microscopy (TEM), X-ray diffraction (XRD) and X-ray photoelectron spectrum (XPS). The results show that all coatings were evidently densified in two hours of heating while the measured conductivities were continuously evolved. The phase composition was found contributed more to the conductivity evolutions than the densification. The conversion of CoO to MnCo$_2$O$_4$ was observed and thus endowed the coatings a conductivity of 40 S/cm. A high fraction of Co$^{3+}$ diffraction peaks, a high amount of Mn$^{2+}$ and a low content of Co$^{2+}$ jointly showed that more Co$^{3+}$ occupied the B site of AB$_2$O$_4$ phase and more Mn$^{2+}$ stood at the A site, indicating a stoichiometric composition of MnCo$_2$O$_4$. Annealing twins were detected by TEM and EBSD for the heated coatings but only a limited contribution to the evolution of the conductivity was considered. Finally, we suggest a high flowrate of plasma gas and a high input energy to prepare spinel coatings with designed conductivity.

**Keywords:** atmospheric plasma spray; spinel coatings; conductivity evolution; densification; spray parameters

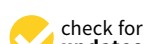



## 1. Introduction

Solid oxide fuel cell (SOFC) is clean power convertor that directly converts chemical energy into electrical energy without particles or NO$_x$ emission [1]. In SOFC power generation system, SOFC stack, which consists of heaped single cells, interconnect, gasket and current collector, also known as repeat unit, is the site where fuel gas is consumed to generate output power at elevated temperature [2,3]. To achieve higher performance and better stability of the SOFC stack, the repeat units should keep good conformity. The interconnect provides physical isolation between two adjacent repeat units while allowing current conduction. Moreover, on the interconnect gas flow channels should be arranged to allow optimized fuel usage. Due to the excellent processing performance and the outstanding resistance to thermal oxidation, ferritic stainless steel is widely used as interconnect material [3–9]. However, continuously exposed to oxidative environment at temperature as high as 800 °C leads to oxidization and evaporation of Chromium which

is one of the most important element of stainless steel. It has been reported that the preliminarily formed $Cr^{3+}$ can be further oxidized to volatile $Cr^{6+}$ at cathode condition, resulting in cathode poisoning and accelerated degradation of performance [10–13].

As a practical solution to the poisoning problems, protective coatings are proposed on the surface of metallic interconnects [14–16]. Although numerous works have been reported to develop protective coatings of different materials, such as $Cu_{1.4}Mn_{1.6}O_4$ [17], $LaCrO_3$ [18,19], $LaCoO_{3-\delta}$ [20], $YCrO_3$ [21], $(La, Sr)CoO_3$ [22], $MnCo_2O_4$ [23] and $Mn_{1.5}Co_{1.5}O_4$ [24], $Mn_xCo_{3-x}O_4$ (0 < x < 3) spinel has been extensively studied due to its high conductivity and excellent performance to restrain Cr volatilization. Various methods were applied to develop superior spinel coatings, such as sol-gel [25], electro-plating [17], slurry coating [26], electrophoretic deposition [27], screen printing [28] and plasma spray [24]. Among these processes only plasma spray can efficiently prepare spinel coating without post heating treatment. However, due to the relatively high temperature of plasma plume, decomposition of $(Mn, Co)_3O_4$ (noted as MnCoO) spinel occurs and thus draws down the conductivity of the obtained coating.

Therefore, for promoting the conductivity of plasma-sprayed spinel coatings, this work was proposed to prepare MnCoO spinel coatings with controlled parameters. By applying micro-morphology characterizations and conductivity measurements the effect of spray parameters on the performance of MnCoO spinel coatings was studied and discussed.

## 2. Methodology

### 2.1. Experimental Setups and the Preparation of Coatings

Atomized $Mn_{1.5}Co_{1.5}O_4$ spinel powder (Terio, Qingdao, China) with average diameter of 28.5 μm ($d_{10}$ = 19.2 μm and $d_{90}$ = 41.3 μm) was employed as the feedstock (see in Figure 1). As shown in Figure 2, the employed $Mn_{1.5}Co_{1.5}O_4$ powders is mainly composed of three phases, $MnCo_2O_4$, $Mn_3O_4$ and $Co_3O_4$. Both $Mn_3O_4$ and $Co_3O_4$ exhibit spinel structure known as the space group of Fd3m.

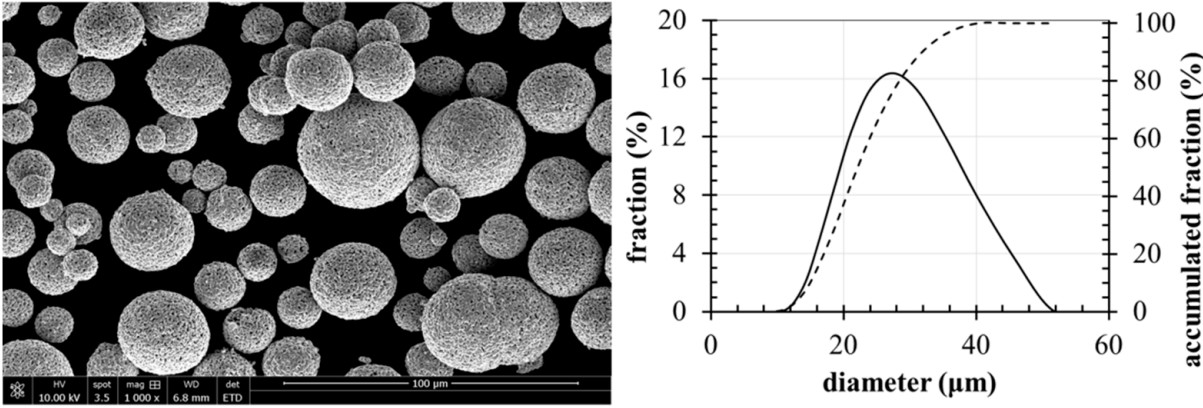

**Figure 1.** The micromorphology and granularity of spinel feedstock.

Spinel coatings were prepared via a F6 torch (Metco, Swiss) installed on a robotic arm. At the exit of the torch nozzle a powder injector with inner diameter of 1.5 mm was vertically mounted. Alumina substrates (Φ18.5 mm × 1.5 mm) were used to allow the measurement of the conductivity of the prepared coatings. For achieving a better adhesion between the coating and the substrate, all substrates were firstly swept by the plasma plume (Ar and $H_2$) to clean and to preheat the substrate to 300 °C. For obtaining coatings with thickness about 30 μm, all samples were sprayed two passes at a traversing speed of 500 mm/s. 4 SLPM Argon was used as the carrier gas to give a powder feeding rate of 14.9 g/min. In addition, the spray distance was maintained at 110 mm. More spray parameters are provided in Table 1.

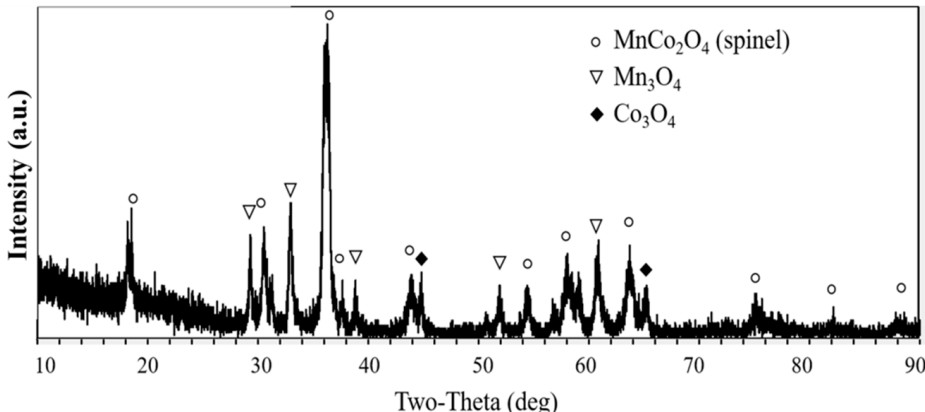

**Figure 2.** X-ray diffractometry patterns of $Mn_{1.5}Co_{1.5}O_4$ spinel powders.

**Table 1.** The applied spray parameters.

| Sets | Current (A) | Voltage (V) | Input Power (kW) | Primary Gas (Ar, L/min) | Secondary Gas ($H_2$, L/min) |
|---|---|---|---|---|---|
| 1 | 450 | 79 | 35 | 50 | 11 |
| 2 | 500 | 79 | 40 | 50 | 11 |
| 3 | 650 | 79 | 50 | 50 | 11 |
| 4 | 650 | 69 | 45 | 40 | 7 |
| 5 | 650 | 84 | 55 | 70 | 7 |

*2.2. Characterization of MnCoO Spinel Coatings*

The crystalline phase and composition of the prepared MnCoO spinel coatings were checked by X-ray diffraction (XRD, Bruker D8 Advance, Bruker Cop., Karlsruhe, Germany) in the range of 10°–90° (0.01°/step, 0.177s/step), and by transmission electron microscope (TEM, JEOL-2100F, Akishima, Japan). Surficial and sectional morphologies of MnCoO spinel coatings were characterized by scanning electron microscope (SEM, FEI Nano450), X-ray photoelectron spectroscopy (XPS, Thermo fisher Scientific Nexsa, Thermo fisher Scientific Inc., Waltham, MA, USA) and Electron Backscattered Diffraction (EBSD). Four-probe method was applied to determine the conductivity of coatings in a tubular furnace [29]. As shown in Figure 3, the obtained samples were firstly placed in the quartz tube of the tubular furnace, then heated up to 700 °C with a heating rate of 5oC/min and held on. A constant current of 0.2 A was applied between pin 1 and pin 4. Voltage between pin 2 and pin 3 was continuously recorded. During the measurement, the recorded voltage was found consecutively decreased and finally stabilized at a constant value. For this reason, samples were held on at the measuring temperature for 15 h to allow a sufficient densification of the spinel coatings. For better understanding, all samples in as-sprayed state were noted as "as"; and samples were marked as "heated" after the conductivity measurement.

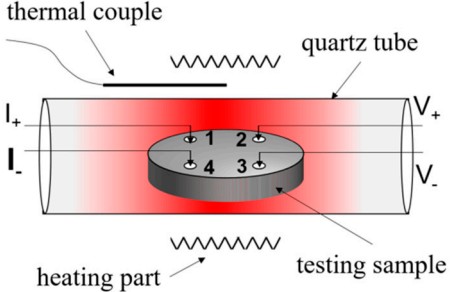

**Figure 3.** The Scheme of experimental setups for conductivity measurement.

## 3. Results and Discussion

### 3.1. Morphology of MnCoO Spinel Coatings

The morphology on the cross section of the as-sprayed MnCoO spinel coatings (marked with "as") are shown in Figure 4. Pores and intervals were found overall the cross section of the five as-sprayed samples. The pores and intervals were considered as the result of a low degree of melting. During the spray, the injected particles were accelerated and consecutively collided with the substrate, resulting lamella structures. For partially melted particles, the momentum gained from the plasma plume was too low to conduct a sufficient flattening that brings gaps or intervals to the coating. As sample No. 1 was prepared with the lowest input power, the obtained coating consisted of visible intervals and pores. Prepared at a moderate input power, sample No. 2 also exhibited a fragmental morphology. As shown in Table 1, maintaining the plasma gas at 61 SLPM but increasing the input power from 35 kW to 40 kW, the in-flight particles absorbed more heat from the plasma plume, thus more particles were well melted and highly softened, leading to a better flattening after the impact and hence less gaps and intervals in the coating as shown by No. 2-as in Figure 4.

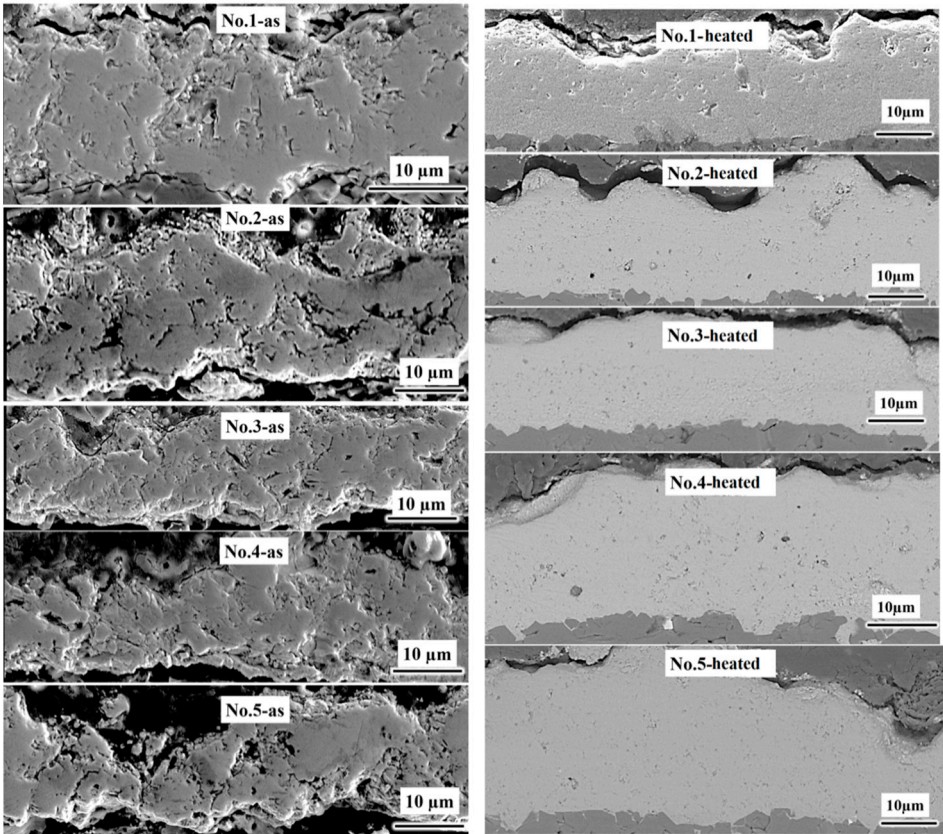

**Figure 4.** The morphology on cross section of MnCoO spinel coatings in the as-sprayed state (marked with "as") and after the conductivity test (marked with "heated").

However, further increasing the input power to 50 kW, the as-sprayed coating No. 3 remained porous with pores and intervals as provided by No. 3—as in Figure 4. Sample No. 4, obtained at 45 kW input power and 47 SLPM plasma gas, presented a fragmental morphology like sample No. 3. By increasing the input power to 55 kW and tuning the plasma gas to 77 SLPM, sample No. 5 shew a worse morphology in the as-sprayed state compared with other samples. For ceramic coatings with poor toughness, such as MnCoO used in this work, pores, intervals or even cracks are inevitable during the preparation process due to the high thermal gradient brought by the plasma plume.

Despite that, a significant densification of coatings was observed after the conductivity measurement at 700 °C. As shown by the heated coatings in Figure 4, pores and intervals found in the as-sprayed coatings were significantly eliminated as the result of densification effects. The densification was conducted during the hold-on at the measuring temperature, which can be considered as an annealing process, that the thermal stress from the plasma spray was released, pores were exhausted and the intervals were healed. Moreover, the densification was also attributed to the absorption of oxygen during the hold-on, which drives the lattice rearrangement and promotes the recrystallization. Resultantly, all samples were obviously densified without visible gaps or intervals, but still with some small pores. The porosity of the spinel coating was measured by image analysis and the value is provided in Table 2.

**Table 2.** Porosity of plasma-sprayed spinel coatings.

| Sample | As-Sprayed/(%) | Heated/(%) |
| --- | --- | --- |
| No. 1 | $8.37 \pm 0.72$ | $3.04 \pm 0.66$ |
| No. 2 | $7.70 \pm 1.10$ | $1.82 \pm 0.65$ |
| No. 3 | $7.80 \pm 1.43$ | $3.05 \pm 0.39$ |
| No. 4 | $9.90 \pm 0.01$ | $2.16 \pm 0.19$ |
| No. 5 | $7.83 \pm 0.84$ | $2.02 \pm 0.66$ |

### 3.2. Electrical Conductivity

The electrical conductivity ($\rho$) of coatings is obtained as $\rho = 1/(c*R*W)$ with $R$ presenting the calculated resistance ($\Omega$), $W$ giving the thickness of coating (cm) and $c$ denoting the coefficient and $\rho$ representing the specific conductivity (S/cm). The obtained conductivity of plasma-sprayed MnCoO spinel coatings is presented in Figure 5 and the value is provided in Table 3. For most of these samples, the conductivity plot consists of three stages: the low-increase-rate stage (stage I), the high-increase-rate stage (stage II) and the stable stage (stage III). Exclusively, sample No. 4 exhibits a progressively increased conductivity during the hold-on with a duration up to 950 min. To achieve stage III, sample No. 1 took about 510 min, sample No. 2 consumed about 300 min, sample No. 3 needed about 630 min and sample No. 5 experienced about 480 min. As shown in Figure 5, at stage III, sample No. 5 possessed the highest conductivity with a value of 40 S/cm and sample No. 3 exhibited a medium conductivity of 28 S/cm. Sample No. 1, sample No. 2 and sample No. 4 shew a relatively low conductivity with a value of 9 S/cm, 12.5 S/cm and 4.5 S/cm, respectively.

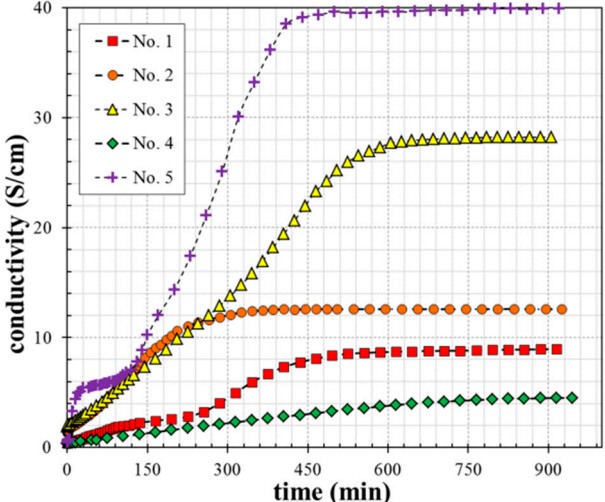

**Figure 5.** Conductivity at 700 °C of plasma-sprayed MnCoO spinel coatings.

**Table 3.** Conductivities of plasma-sprayed spinel coatings.

| Sample | Conductivity in Stage I | Conductivity in Stage II | Conductivity in Stage III |
| --- | --- | --- | --- |
| No. 1 | 3 S/cm | 9 S/cm | 9 S/cm |
| No. 2 | 6 S/cm | 12 S/cm | 12.5 S/cm |
| No. 3 | 14 S/cm | 28 S/cm | 28 S/cm |
| No. 4 | 2 S/cm | 4 S/cm | 4.5 S/cm |
| No. 5 | 10 S/cm | 39 S/cm | 40 S/cm |

As shown in Figure 4, holding on at 700 °C brought a significant densification to the coatings that the number of interfaces, gaps and pores was significantly diminished. The electronic resistance arose from grain boundaries was thus dramatically decreased that electrons hoped between grains with less energy consumption. As a result, the conductivity was continuously increased. However, due to the densification happens quickly and completed in a short time (see Figure A1 in Appendix A), the increase of conductivity after 2 h of hold-on was induced by phase transformations.

In a magnified view of the morphologies of a sample heated for different durations, two distinguishable phases were found as Mn-rich phase and Co-rich phase (see Figure A2 in Appendix A). The Mn-rich phase is in dark gray and the Co-rich phase is in gray or light gray. The element fraction of these phases was also provided in the complementary file (see in Table A1). Therefore, it is true that during the conductivity measurement, the plasma-sprayed coatings were firstly densified within 2 h (stage I), then the Mn-rich phase and Co-rich phase were slowly diffused to each other and the excess elements were expelled to the surface forming co-rich traces at about 5 h, then the traces turned to sweat-like precipitations during stage II; and during stage III the adjacent sweat-like precipitations were merged to conduct a consecutively growing up. Moreover, as shown in Figure 5 the increase of the measured conductivity of the plasma-sprayed MnCoO spinel coatings was mainly happened in stage II. Therefore, it is believed that the increase of the conductivity was mainly attributed to the phase transformation during stage II.

### 3.3. Phase Composition

Moreover, the evolution of the conductivity in stage II was associated with the variation of phase compositions according to the XRD patterns provided in Figure 6. The XRD patterns for as-sprayed and heated coatings were displayed in Figure 6. The presence of $Al_2O_3$ was due to the penetration of X-ray to the substrate. Diffraction peaks indicate the existence of $MnCo_2O4$ and $CoO$ for the as-sprayed samples, while only spinel phase is detected after the conductivity measurement. As the raw material consists of spinel phases, $Co_3O_4$ and $Mn_3O_4$, the presence of $CoO$ is considered as a result of the decomposition of $Co_3O_4$ during the plasma spray [30]. $CoO$ is a P-type semi-conductor with a band gap of 2.6 eV, showing relatively low conductivity compared to $MnCo_2O_4$ phase. During the conductivity measurement, samples were exposing to in hot air (700°C) that the thermal energy and oxygen were continuously provided to initiate the conversion of $CoO$ to $MnCo_2O_4$ spinel, bringing significant improvement to the conductivity.

### 3.4. Elemental Composition

For further understanding the mechanism of the conductivity evolution, both sample No. 4 (with the lowest conductivity) and sample No. 5 (with the highest conductivity) were checked by XPS in as-spray state and after the conductivity measurement. As shown in Figure 7a, Co, Mn, O and Au are identified. The presence of Au is due to the plating before SEM observations. In Figure 7b, the high-resolution Co 2p spectrum of MnCoO is presented and Co $2p_{3/2}$ (780.1 eV) and Co $2p_{1/2}$ (795.4 eV) can be identified with energy gap of 15.3 eV [31]. The coexistence of $Co^{3+}$ and $Co^{2+}$ in the spinel coatings is confirmed by the satellites at 786.2 eV and 802.6 eV [32]. The high-resolution spectra of Mn 2p in Figure 7c exhibits two broadened profiles with peaks located at 640.6 eV and 651.9 eV

which are respectively assigned to Mn $2p_{1/2}$ and Mn $2p_{3/2}$ [33]. In Figure 7d the spectrum of O 1s performs two peaks at 529.8 eV and 531.5 eV. The peak located at 529.8 eV is corresponded to the typical metal-oxygen bond of the lattice oxygen in $MnCo_2O_4$ spinel and the peak at 531.5 eV is attributed to the absorbed oxygen groups (mainly O–H) [34]. Moreover, by performing a deconvolution of these high-resolution spectra, the proportion of $Co^{2+}$, $Co^{3+}$, $Mn^{2+}$, $Mn^{3+}$ and $Mn^{4+}$ can be obtained and provided in Table 4.

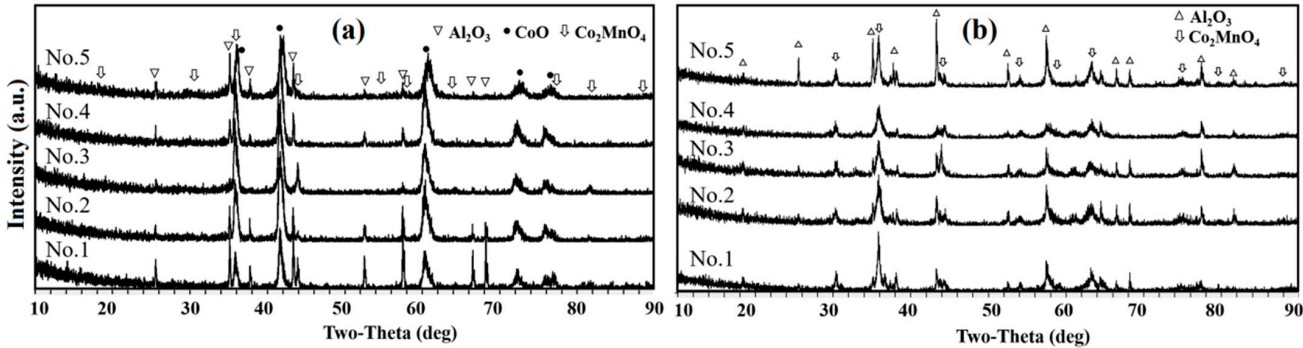

**Figure 6.** XRD patterns of MnCoO spinel coatings: (**a**) as-sprayed and (**b**) heated.

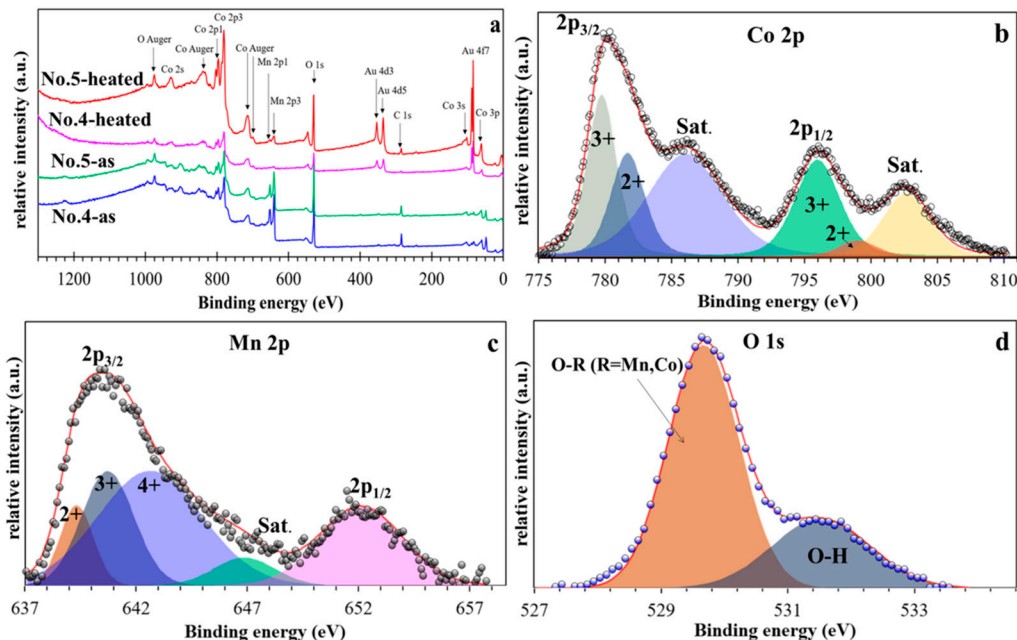

**Figure 7.** XPS spectra of MnCoO spinel coatings: (**a**) survey (LP: low), (**b**) Co 2p of sample No. 5, (**c**) Mn 2p of sample No. 5 and (**d**) O 1s of sample No. 5.

**Table 4.** The areal proportions of elements for as-sprayed and heated samples.

| Elements | Proportions of As-Sprayed Samples (%) | | Proportions of HEATED SAMPles (%) | |
|---|---|---|---|---|
| | No. 4 | No. 5 | No. 4 | No. 5 |
| $Co^{3+}$ | 72.65 | 69.47 | 47.43 | 82.59 |
| $Co^{2+}$ | 27.35 | 30.52 | 52.57 | 17.41 |
| $Mn^{2+}$ | 14.27 | 13.56 | 1.90 | 13.00 |
| $Mn^{3+}$ | 38.04 | 45.17 | 57.34 | 29.04 |
| $Mn^{4+}$ | 47.69 | 42.27 | 40.76 | 57.96 |

Areal proportions of elements were found variated between samples during the conductivity measurement, as provided in Table 4. It is well accepted that spinel phase can be described as $AB_2O_4$ with A denoting transition metallic element in 2+ valence state and B presenting transition metallic element in 3+ state. In this work, the spinel phase $MnCo_2O_4$ with $Mn^{2+}$ occupying A site and $Co^{3+}$ settling at B site exhibits an theoretic conductivity with a value as high as 60 S/cm. With a higher proportion of $Co^{3+}$ and a lower proportion of $Co^{2+}$ the heated sample No. 5 is considered with a high fraction of conductive spinel phase and thus a higher conductivity. What's more, the fraction of $Mn^{2+}$ of the as-sprayed sample No. 4 and No. 5 indicates that the heated sample No. 4 less Mn presents at A site and as a result the detected $MnCo_2O_4$ spinel phase is less. Although, Mn can occupies the B site of an inverse spinel, the valance will be remained as 2+ that the above-mentioned variations of the elemental proportions can be always observed by XPS. Therefore, it is reasonable to believe that the low conductivity of sample No. 4 is determined by the low proportion of $MnCo_2O_4$ spinel phase and the high conductivity of sample No. 5 is originated form the high proportion of $MnCo_2O_4$ spinel phase in the coating.

As shown in Figure 5, the measured conductivity was finally stabilized at 40 S/cm for sample No. 5, at 4.5 S/cm for sample No. 4, at 28 S/cm for sample No. 3, at 12.5 S/cm for sample No. 2 and at 9 S/cm for sample No. 1. Once an equilibrium was achieved, the phase transformation was then stopped and hence a constant conductivity was established. The conductivity plots of the spinel coatings thus turned to a stable stage, stage III.

*3.5. The Microstructure of Sliced Samples*

To further check the microstructure and the coating composition after the conductivity measurement, slices of measured coating No. 4 and No. 5 were prepared by FIB (focused ion beam) as shown in Figure 8a,b. For both samples all grains are well bonded together without visible gaps or intervals or cracks. The measured mean grain size of both samples was 0.7 μm and 0.65 μm respectively (more details are provided in Appendix A Figure A4). A well bonded boundary can bring great contributions to promote the hopping of free electrons. Moreover, according to Figure 8c,d, both samples were found with visible annealing twins extending from grain boundaries to the grain interior. More details were provided in Figure A3.

The presence of annealing twins can further diminish the boundary resistance. Since the adjacent grains of annealing twins exhibit highly consistent lattice structures, less energy will be consumed for free electrons to traverse these grains. Sample No. 4 was found with more annealing twins of high degree boundaries than that of sample No. 5 as marked by red lines in Figure 8c,d. The development of annealing twins can yield sub-grains as shown in Figure 8e,f. A large sub-grain can be observed in Figure 8e with size of 10 μm, inside which numerous small degree grain boundaries are observed. However, for highly conductive sample No. 5 the observed sub-grains are smaller than 3 μm as shown in Figure 8f and the average size of sub-grains of sample No. 5 is visibly smeller than that of sample No. 4. Therefore, it is believed that the size of grains or sub-grains brings a limited contribution to the conductivity difference between sample No. 4 and No. 5. For better understanding the conductivity evolution and the difference between samples, EDS was applied at selected spots as shown in Figure 8a,b.

The obtained element fractions are listed in Table 5. For heated sample No. 4 the mean percentage of Mn and Co is 25.26% and 29.71%, respectively. But for heated sample No. 5 the percentage of Mn and Co at spot 1, spot 2 and spot 3 is close to 18%, while at spot 4 and spot 5 the content of Mn becomes about 25% and Co is about 28% which is like that of heated sample No. 4. In average, the percentage of Mn and Co of sample No. 5 is 20.82% and 21.80% respectively. It shows an atomic ratio of 1:1.05:2.76 (Mn:Co:O) and thus signifies a composition of $Mn_{1.5}Co_{1.5}O_4$ for sample No. 5. Meanwhile, the atomic ratio of sample No. 4 is 1:1.18:1.78 (Mn:Co:O) indicating a deoxidized spinel phase, that explains the conductivity difference between sample No. 5 and sample No. 4 as provided in Figure 5.

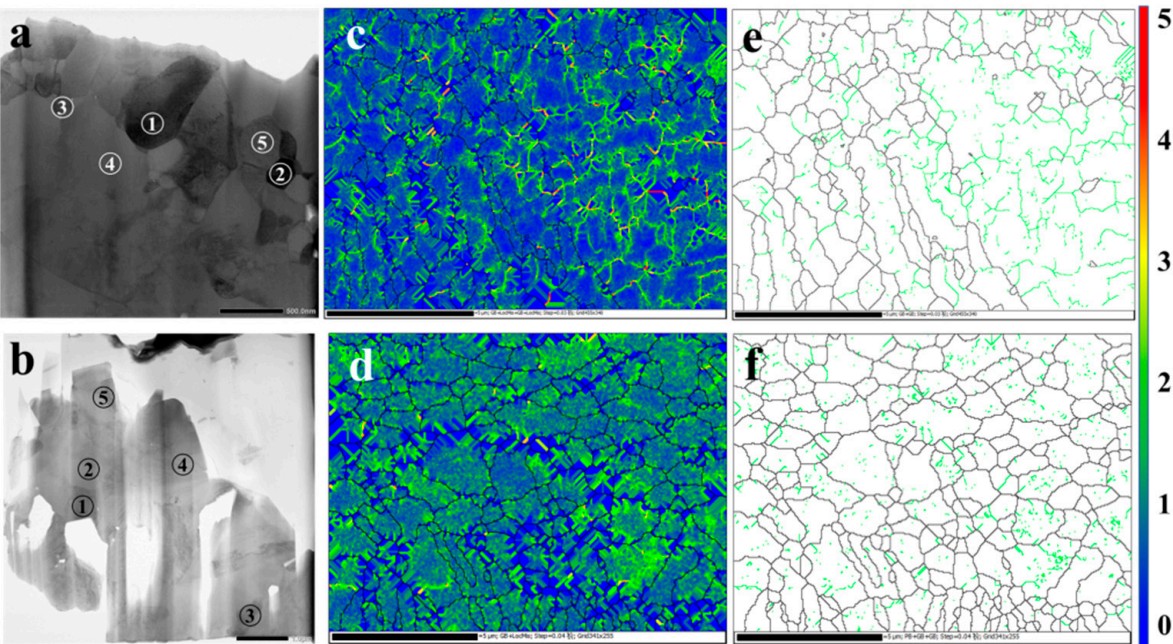

**Figure 8.** Micromorphology of samples after conductivity measurement: (**a,b**) microstructures of sliced samples prepared by FIB; (**c,d**) local mis-orientation of samples; (**e,f**) grain boundaries of samples; (**a,c,e**) denote sample No. 4; (**b,d,f**) represent sample No. 5. The grain boundaries and local mis-orientation were obtained by EBSD.

**Table 5.** Element percentages of heated sample No. 4 and No. 5.

| Percentage (at.%) | | Spot 1 | Spot 2 | Spot 3 | Spot 4 | Spot 5 | Average | Mn:Co:O |
|---|---|---|---|---|---|---|---|---|
| | Mn | 26.68 | 25.77 | 23.75 | 25.17 | 24.93 | 25.26 | 1 |
| Sample No. 4 | Co | 30.39 | 30.31 | 30.11 | 27.96 | 29.76 | 29.71 | 1.18 |
| | O | 42.93 | 43.92 | 46.14 | 46.87 | 45.31 | 45.03 | 1.78 |
| | Mn | 18.23 | 17.58 | 17.39 | 25.27 | 25.65 | 20.82 | 1 |
| Sample No. 5 | Co | 19.18 | 16.99 | 17.81 | 26.82 | 28.21 | 21.80 | 1.05 |
| | O | 62.59 | 65.43 | 64.8 | 47.91 | 46.14 | 57.37 | 2.76 |

*3.6. Effects of Spray Conditions on Coatings*

The above discussions systemically investigated the micromorphology, elemental composition, phase component, boundary state and grain size between samples prepared at different parameters, but for a deeper comprehension to this work, the effects of spray conditions on spinel coatings are described in this section. As sample No. 4 was prepared at a low flowrate of plasma gas and at a moderate input energy, spinel particles injected to the plasma plume were quickly heated and melted to form droplets. At the same time, deoxidization happened due to the high temperature and the low plasma velocity. According to the phase diagram of $Mn_3O_4$-$Co_3O_4$ [30], both spinel phase and (Co, Mn)O phase presented in the droplets. Differently, sample No. 5 was prepared at a higher flowrate of plasma gas and a higher input energy that particles injected into the plasma plume were shortly accelerated to a velocity much higher than that achieved during the preparation of sample No. 4. The higher the velocity is for the injected particles, the shorter the time exposing in the hot plasma plume is. With a shorter exposing in the hot plasma plume, the deoxidization of spinel phase is in some degree interrupted, thus more conductive spinel phases are remained to provide the obtained coatings a higher conductivity.

It can be inferred that for sample No. 1, No. 2 and No. 3, the amount of spinel phases maintained in the coating is between that of sample No. 4 and that of sample No. 5 according to the measured conductivities shown in Figure 5. As mentioned in Section 3.1, sample No. 1, No. 2 and No. 3 were prepared at a constant flowrate of plasma

gas (61 L/min) with a controlled input energy: sample No. 1 was prepared as the lowest input energy, sample No. 2 was prepared at a moderate input energy and sample No. 3 was obtained at a higher input energy. A variation of the input energy brings significant difference in heating condition of particles.

For sample No. 1, due to the low input energy, most of the particles injected to the plasma plume were unable to be completely melted. It means particles may be partially melted to form core-shell structure that solid core is surrounded by liquid shell. According to the phase diagram of $Mn_3O_4$-$Co_3O_4$ [30], there is a wide (Co, Mn)O area with temperature ranged from 1000 °C to 1700 °C. Unfortunately, the temperature of theses partially melted particles is often located in this region that the raw spinel phase was primarily turned to the low conductive phase (Co, Mn)O. By employing a higher input energy, such as 40 kW of sample No. 2, the plasma temperature was improved and the molten state of particles was ameliorated, the generation of (Co, Mn)O was slightly restrained that brings a slight augmentation in the conductivity as shown in Figure 5. By arising the input energy to 50 kW (sample No. 3), more injected particles could be heated to a temperature higher than 1700 °C and thus be completely melted. After the impacting and flattening, the melted particles were quickly solidified on the 'cold' substrate, that the high-temperature phase was retained and only a limited amount of (Co, Mn)O was generated. As a result, the conductivity of sample No. 3 was much higher than that of sample No. 1 and sample No. 2, but lower than that of sample No. 5. Thus, a high plasma temperature and a high plasma velocity is favorable to prepare highly conductive spinel coatings.

## 4. Conclusions

In summary, MnCoO spinel coatings were obtained by atmospheric plasma spray with controlled parameters: the flowrate of plasma gas and the input energy. The as-sprayed coatings were found with gaps and intervals which were considered originated from the preparation process. By performing conductivity measurements at 700°C an evident densification was observed over all samples that the gaps and intervals were healed. The densification caused significant improvement of the electrical conductivity of the obtained spinel coatings. A typical four-probe method was applied to perform the measurement of conductivity. The obtained conductivity plots exhibited three stages: stage I (the increasing rate of conductivity is low), stage II (the increasing rate is high) and stage III (the conductivity becomes constant). In stage I, the conductivity was considered mainly dominated by the densification effect which decreased the number of boundaries and thus yielded a lower resistance to electron hopping; in stage II, the generation of highly conductive spinel phases contributed to the increase of the measured conductivity; once the generation of conductive spinel phases was completed, the measured conductivity became stable as shown in stage III.

The generation of conductive spinel phases was confirmed by XPS and TEM-EDS. The areal proportion of Mn ions and Co ions obtained by deconvolution of the high-resolution XPS spectra indicates that the measured conductivity in stage II is mainly attributed to $MnCo_2O_4$, the highly conductive spinel phase. Moreover, the sample with a high conductivity was found with an atomic ratio of 1:1.05:2.76 (Mn:Co:O), which signifies a stoichiometric composition of $Mn_{1.5}Co_{1.5}O_4$ known as a mixture of $MnCo_2O_4$ and $CoMn_2O_4$; while sample exhibiting a low conductivity was found with an atomic ratio of 1:1.18:1.78 (Mn: Co: O) which indicates a deoxidized spinel phase.

The variation of stoichiometric composition of the obtained coatings can be understood from the preparation process. By applying a high plasma temperature and a high plasma velocity, the molten degree of particles was improved and thus more particles were completely molten. Moreover the exposing time of particles in the hot plasma was limited that the generation of low conductive phases (Co, Mn)O was restricted and thus more high-temperature phases were retained but with some defects. During the conductivity measurements, the high-temperature phases were gradually healed and turned back to

highly conductive spinel phases. Therefore, a high flowrate of plasma gas and a proper high input power are suggested to prepare spinel coatings with high conductivity.

**Author Contributions:** Formal analysis, J.Z.; Investigation, J.Z. and K.W.; Methodology, C.S. and T.L.; Project administration, M.L.; Resources, C.D.; Supervision, C.S. and T.L.; Writing—original draft, J.Z.; Writing—review & editing, T.L. and C.Y. All authors have read and agreed to the published version of the manuscript.

**Funding:** This research was funded by the National Key R&D Program of China grant number 2018YFB1502603, Guangdong Academy of Science grant number 2019GDASYL-0102007, and the Foreign Cooperation Platform Project of Guangdong Province grant number 2020A050519001.

**Institutional Review Board Statement:** Not applicable.

**Informed Consent Statement:** Not applicable.

**Data Availability Statement:** The data presented in this study are available on request from the corresponding author.

**Conflicts of Interest:** The authors declare no conflict of interest. The founding sponsors had no role in the design of the study; in the collection, analyses, or interpretation of data; in the writing of the manuscript, and in the decision to publish the results.

**Appendix A**

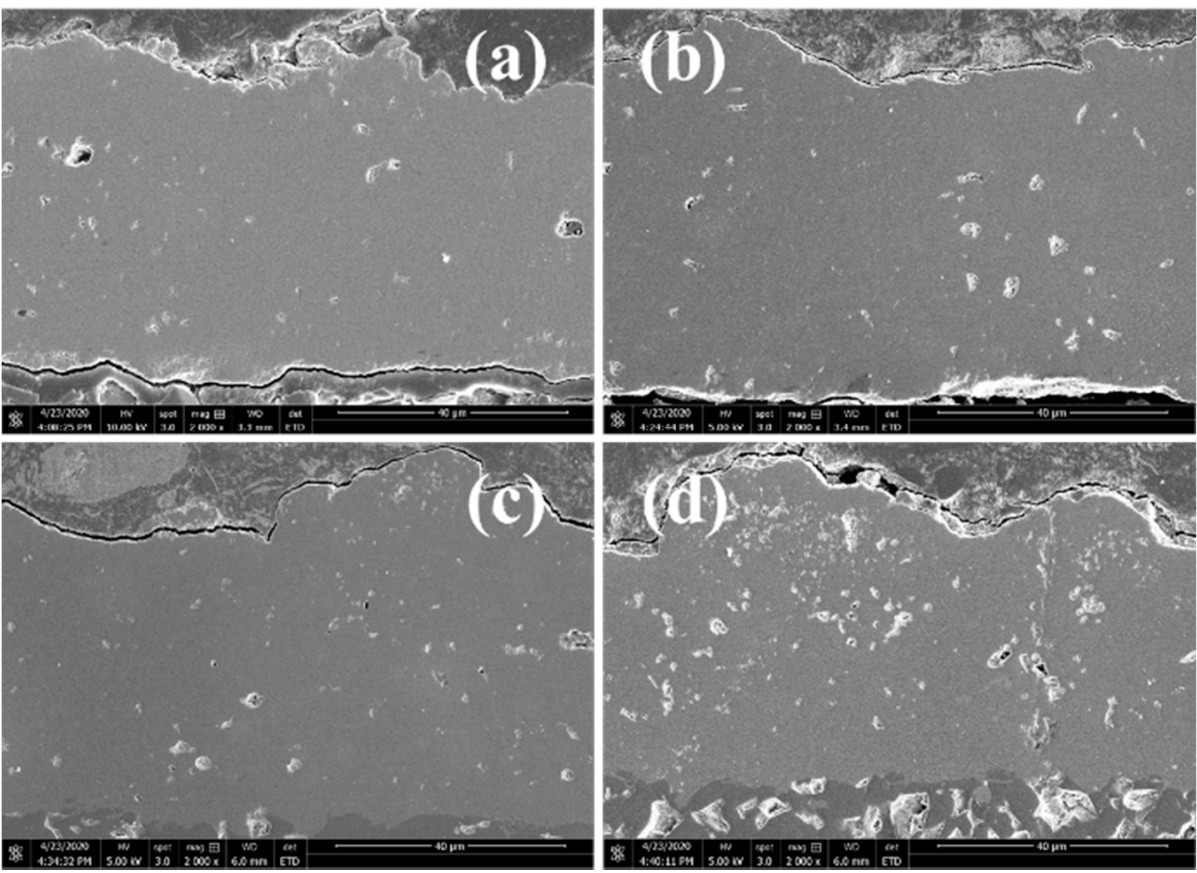

**Figure A1.** Densified morphologies of plasma-sprayed MnCoO coatings held at 700 °C in air: (**a**) 2 h, (**b**) 5 h, (**c**) 11 h and (**d**) 20 h.

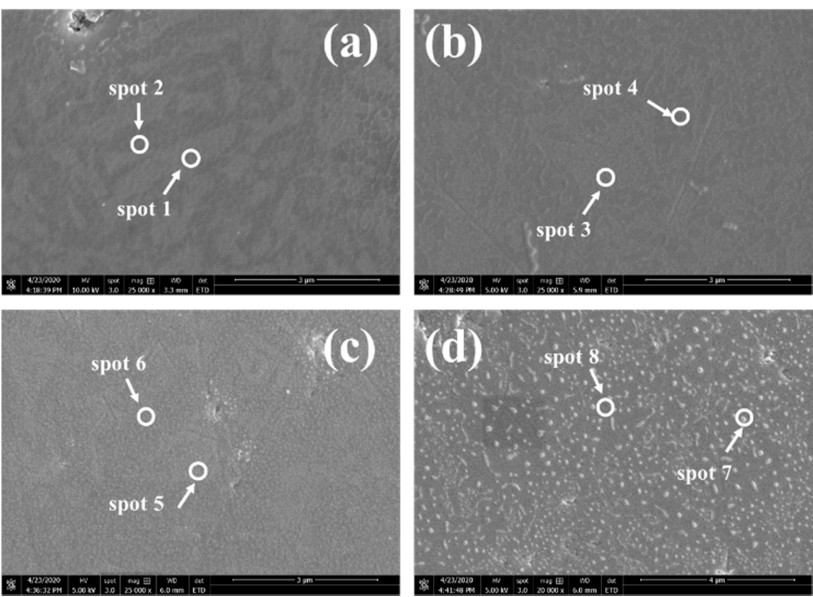

**Figure A2.** Magnified morphologies of plasma-sprayed MnCoO coatings held at 700 °C in air: (**a**) 2 h, (**b**) 5 h, (**c**) 11 h and (**d**) 20 h.

**Table A1.** Atomic fraction of Mn, Co and O in the plasma-sprayed MnCoO coatings.

| at.% | 2 h | | 5 h | | 11 h | | 20 h | |
| --- | --- | --- | --- | --- | --- | --- | --- | --- |
| | Spot 1 | Spot 2 | Spot 3 | Spot 4 | Spot 5 | Spot 6 | Spot 7 | Spot 8 |
| Mn | 29.12 | 51.67 | 3.05 | 16.48 | 0.03 | 14.60 | 4.81 | 13.70 |
| Co | 24.04 | 3.97 | 28.78 | 3.52 | 39.27 | 5.00 | 23.02 | 6.82 |
| O | 46.84 | 44.35 | 68.17 | 80.00 | 61.33 | 80.40 | 72.17 | 79.48 |

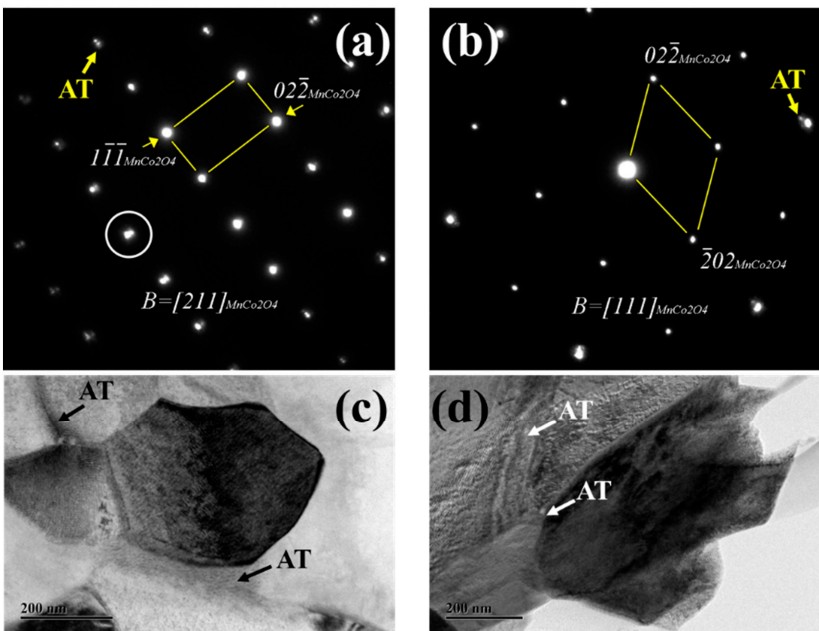

**Figure A3.** the high-resolution transmission electron microscopy and the selected area electron diffraction of the annealed samples: (**a**,**c**) No. 4, (**b**,**d**) No. 5. Both samples were identified with $MnCo_2O_4$ spinel phase and the presence of annealing twins which marked as "AT" in the figures.

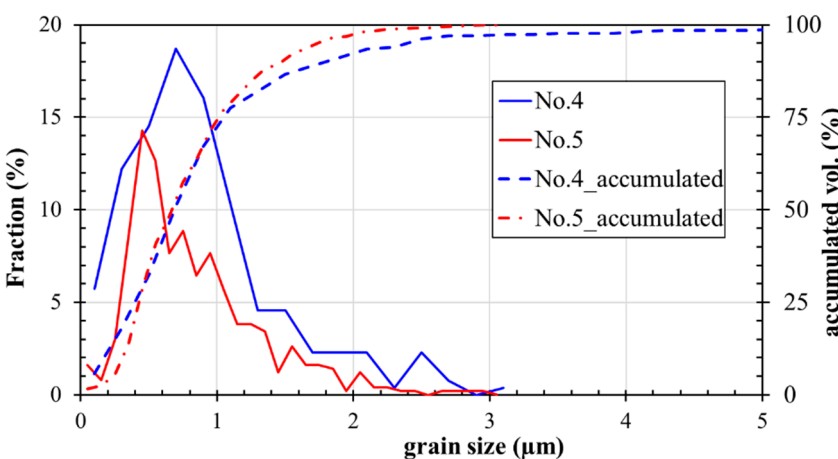

**Figure A4.** the grain distribution of heated sample No. 4 and No. 5.

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
