# Peer review of "The Microstructure and Conductivity Evolution of Plasma-Sprayed (Mn, Co)3O4 Spinel Coatings during Conductivity Measurements at Elevated Temperature"

_coatings, doi:10.3390/coatings11050533_

Round 1

Reviewer 1 Report

The reviewed article deals with microstructure and conductivity evolution on plasma sprayed (Mn, Co)3O4 spinel coatings during conductivity measurements at elevated temperature. The paper is well organized and gives realable and interesting results. Nevertheless there are some remarks:

  1. Feedstock powder - please add d10 and d90 values.
  2. What was the spray distance? It was fixed or variable?
  3. XRD analysis - please add step and time for step.
  4. Caption for Figure 3 - please use "scheme" instead "illustration".
  5. Figure 4 - for "as-sprayed" coatings there is a poor interface between coating and substrate. Please explain it.
  6. Why Authors did not carry out porosity measurements? At least image analysis and compare it before and after conductivity tests.
  7. For EBSD there is no information about grain size, please complete it.

Author Response

I have dealt with the reviewer’s comments seriously, please download the attachment to view the specific response and revise.

Reviewer 2 Report

In the manuscript, the authors have prepared 5 samples of MnCoO spinel layer on alumina substrate by plasma spraying of Mn1.5Co1.5O 4 powder. They varied plasma power, gas flow rate and gas composition. They analysed the influence of preparation conditions and post-deposition annealing on the structural and electrical properties of prepared samples.

The manuscript is well structured and the content is expressed clearly and in proper English. However, the analysis was not done systematically. For some samples plasma power, flow rate and the ratio of Ar and H2 in gas mixture was varied at the same time which not permit to unambiguously conclude about single parameter. Only for the first three samples gas composition and flow rate were kept constant and only plasma power varied. I’m suggesting to do additional experiments with fixed power and varied only one of flow rate or Ar/H2 ratio.

Also, the authors for several Figures (5, 7), in the Discussion are focused on the repetition of numerical data presented in figures avoiding deeper discussion of results.

Authors mentioned and suggested that the material densification effect during post-deposition annealing is related to change in structure and electrical properties. To stronger support this statement I’m suggesting to quantify the change in density by proper additional experiments.

Below is a short list of additional questions, comments and suggestions that can improve the quality of the manuscript:

  • Why the spinel powder used for deposition is called Mn1.5Co1.5O4 if it consists of MnCo2O4, Mn3O4 and Co3O4?

  • Fig 2: There are several additional XRD peaks that are not indexed and labelled. For example at 51°, 57°, 59° etc.

  • Line 80-81: What kind of plasma was used for substrate cleaning? Please specify!

  • Fig 5: In the text is repeated information presented in the figure. It’s better to put values in a table and in the text leave only discussion of the obtained results.

  • Fig 5. It’s not clear when the conductivity measurement is started? At the same time when heating was started (at room temperature) or when 700 C deg was reached? If the electrical measurement was started at room temperature then also temperature influence on Phase I should be considered.

  • Fig 7: In legend for a) subfigure mentioned sample No. 8???

  • Fig 7: Comment same as for Fig 6. In the text are only repeated values presented in Table 2.

Author Response

(The authors gave the same response as above.)

Round 2

Reviewer 1 Report

All my remarks have been included.

Author Response

The reviewer’s comments have been replied, please see the attachment for details.

Reviewer 2 Report

The authors have responded only to minor comments and question correcting the manuscript only partially. They missed to answer the most important questions and comments related to the more systematic selection of the sample preparation parameter set and discussion of density/porosity influence which should be supported by additional experiments as it was suggested in both review reports.

Author Response

(The authors gave the same response as above.)

Round 3

Reviewer 2 Report

The authors have provided answers to all additional questions and modified the manuscript accordingly.